# Paediatric Rational Prescribing: A Systematic Review of Assessment Tools

**DOI:** 10.3390/ijerph17051473

**Published:** 2020-02-25

**Authors:** Fenella Corrick, Sharon Conroy, Helen Sammons, Imti Choonara

**Affiliations:** 1Division of Medical Sciences & Graduate Entry Medicine, University of Nottingham, Royal Derby Hospital Centre, Uttoxeter Road, Derby DE22 3DT, UK; sharon.conroy@nottingham.ac.uk (S.C.); helen.sammons@nhs.net (H.S.); imti.choonara@nottingham.ac.uk (I.C.); 2North Devon District Hospital, Raleigh Park, Barnstaple EX31 4JB, UK

**Keywords:** rational prescribing, paediatrics, rational use of medicines

## Abstract

Rational prescribing criteria have been well established in adult medicine for both research and quality improvement in the appropriate use of medicines. Paediatric rational prescribing has not been as widely investigated. The aims of this review were to identify and provide an overview of all paediatric rational prescribing tools that have been developed for use in paediatric settings. A systematic literature search was made of MEDLINE, Embase, CINAHL and IPA from their earliest records until July 2019 for all published paediatric rational prescribing tools. The characteristics of the tools were recorded including method of development, types of criteria, aspects of rational prescribing assessed, and intended practice setting. The search identified three paediatric rational prescribing tools: the POPI (Pediatrics: Omissions of Prescriptions and Inappropriate Prescriptions) tool, the modified POPI (UK) tool, and indicators of potentially inappropriate prescribing in children (PIPc). PIPc comprises explicit criteria, whereas POPI and the modified POPI (UK) use a mixed approach. PIPc is designed for use in primary care in the UK and Ireland, POPI is designed for use in all paediatric practice settings and is based on French practice standards, and the modified POPI (UK) is based on UK practice standards and is designed for use in all paediatric practice settings. This review describes three paediatric rational prescribing tools and details their characteristics. This will provide readers with information for the use of the tools in quality improvement or research and support further work in the field of paediatric rational prescribing.

## 1. Introduction

Rational prescribing has been defined by the World Health Organisation as “when patients receive the appropriate medicines, in doses that meet their own individual requirements, for an adequate period of time, and at the lowest cost both to them and the community” [1]. It has been poorly studied in paediatric patients; a field that has been described as “an evidence based desert” [2]. Irrational prescribing has wide-ranging impacts, from adverse drug reactions and progression of inappropriately managed disease to additional system healthcare costs and antimicrobial resistance. The use of criteria lists as tools to quantify rational prescribing in adult medicine is well established [3]. There are a number of potential benefits to rational prescribing tools; assessment tools enable quantification of the quality of prescribing, which facilitates research into interventions aiming to improve prescribing and allows prescribing in different settings to be compared. This facilitates deeper research into root causes of problematic prescribing or excellent prescribing and fosters collaboration between different groups.

A 2014 systematic review of rational prescribing tools for adults by Kaufmann et al. identified 46 published tools [4]. Of these, 22% did not have a stated target population, while 78% were specifically targeted to prescribing for older adults. Older adults have been identified as a group vulnerable to irrational prescribing due to a variety of factors, including the frequent existence of co-morbidities, polypharmacy, care taking place in a number of different settings, and the effect of ageing on the selection of appropriate medications [5]. Many similar challenges exist in paediatric medicine, with well-recognised developmental changes in physiology and metabolism having a significant impact on pharmacokinetics in children of different ages [6]. In addition, children may receive medications in a number of different settings, including general practice, undifferentiated emergency departments, walk-in centres, paediatric wards in district general hospitals, and specialist paediatric hospital settings. This means that prescribers with varied levels of paediatric experience and expertise may be responsible for prescribing.

Kaufmann et al. explicitly excluded tools targeted to children in their 2014 review. The aim of this review was to carry out a systematic review of paediatric rational prescribing tools in order to produce a comprehensive overview of current tools available to measure rational prescribing in children. This will hopefully facilitate others studying this area.

## 2. Methods

The systematic literature search was designed to identify articles describing tools to assess rational (or inappropriate) prescribing for children.

Inclusion criteria were: articles describing tools targeted at evaluating the rationality or appropriateness of prescriptions for children (aged less than or equal to 18 years), updated and revised versions of previously published tools, and including tools limited to specific drugs, drug groups, diseases or disease groups.

Exclusion criteria were: tools targeting adults, tools without specified target patient groups, indicators that assess rates or percentages of prescription types in a population, articles describing a validation study of a previously published tool, educational interventions aimed at improving prescribing, and guidelines describing recommended prescribing.

### Search Strategy

The search was conducted in four databases in order to attempt to capture relevant medical, nursing and pharmaceutical research. These were: Ovid MEDLINE, Embase, International Pharmaceutical Abstracts (IPA) and CINAHL. Databases were searched from their earliest records possible until the start of July 2019.

Search terms to capture studies including children were derived from the recommended search strategy described by Kastner et al. 2006 [7], as these have demonstrated high sensitivity. The MeSH term “inappropriate prescribing” was introduced in 2011, and was previously incorporated in the broad term “Drug therapy”. Search terms for rational prescribing were derived from the systematic review of adult rational prescribing tools by Kaufmann et al. 2014 [4]. The combined terms were:

(inappropriate prescribing or suboptimal prescribing or inappropriate medication or inappropriate practices or drug prescriptions or Medication Appropriateness Index) and (child* or children* or p*ediatric* or infant* or adolescent*).

All potentially relevant publications were screened by title and abstract and articles that met the exclusion criteria were excluded. The remaining articles were retrieved in full. Full-texts were examined by FC and a second researcher who performed independent full-text screening, independently assessing articles according to the inclusion and exclusion criteria. After this process, any articles without consensus were resolved by discussion and mutual agreement. A manual search of the bibliographies of included texts was completed.

Included articles were analysed by FC to extract the development process and characteristics of the rational prescribing tool.

## 3. Results

The search produced 2142 potentially relevant publications. 234 duplicated articles were removed. 1908 articles were screened by title and abstract and 1736 were excluded (Figure 1). One hundred seventy-two articles were selected for full-text review by two reviewers of which 163 were excluded. The excluded articles screened at full-text did not meet the inclusion criterion of describing rational prescribing tools. Four full-texts were unavailable online from University library resources, and from the British Library. In the case of the four full-texts that were unavailable the abstracts suggested that these articles would not meet the inclusion criteria, although this could not be determined with certainty. Five articles met the inclusion criteria. Bibliography mining of the included articles did not identify any further relevant articles. 

### 3.1. Rational Prescribing Tools Identified

In total, five relevant articles were identified, relating to three paediatric rational prescribing tools. These are shown in Table 1.

Three relevant articles were identified relating to a single tool: POPI (Pediatrics: Omissions of Prescriptions and Inappropriate Prescriptions) tool [9,10,11]. All three included articles are very similar, two in French and one in English, and describe the process of developing the POPI tool. The earliest, from 2011, is a letter describing the tool, giving nine examples of the gastro-intestinal criteria. The 2014 and 2016 articles are English and French language, respectively. They report the consensus validation of the tool and give full details of the criteria. The number of criteria listed differs due to the combination of several criteria together in the latter publication. Note that the 2014 article states there are 104 criteria but lists 105. For the purposes of clarity, from this point reference to the POPI criteria is specifically to the wording and numbering in the English 2014 publication unless otherwise stated.

The other two relevant articles relate to two additional rational prescribing tools. One of these describes the modified (UK) POPI tool [12], a modification of the above POPI tool for application to use in the UK published by the authors of this review in 2017. The other relates to the development of a rational prescribing tool for the evaluation of paediatric prescribing in primary care, indicators of potentially inappropriate prescribing in children (PIPc) [13]. This was developed in Ireland and the UK and published in September 2016.

### 3.2. Characteristics of the Identified Paediatric Rational Prescribing Tools 

All three tools are examples of explicit or mixed rational prescribing tools and are comprised of a number of explicit criteria defining potentially inappropriate prescriptions (PIPs) and potentially inappropriate omissions (PIOs). Both POPI and the modified POPI (UK) also contain some criteria with implicit features. 

The POPI tool comprises 105 criteria (80 PIPs and 25 PIOs) categorised by the authors according to broadly grouped clinical conditions: diverse illnesses, digestive problems, ENT-pulmonary problems, dermatological problems, and neuropsychiatric disorders. The groups are further subdivided into particular symptoms or conditions. The criteria cover a range of aspects of inappropriate prescribing, including overprescribing, underprescribing, and almost all areas of misprescribing except drug-food interactions. 

The modified (UK) POPI tool comprises 80 criteria under the same categories as the original POPI tool, except for the removal of one subcategory (Mosquitos [sic]). The criteria include 60 PIPs and 20 PIOs across the same aspects of inappropriate prescribing as the original POPI criteria.

PIPc comprises twelve criteria of potentially inappropriate prescribing or omissions, categorised according to four physiological systems: respiratory, gastrointestinal, dermatological, and neurological. Seven criteria describe PIPs with potential overprescribing or misprescribing practices, five relate to PIOs.

The characteristics of the identified tools are summarised in Table 2, where bullet points identify aspects of irrational prescribing that are covered by each tool.

### 3.3. Development of the Popi Tool

The methodology used to develop the POPI tool was designed to closely match the development of the STOPP/START criteria, according to the authors [10]. The STOPP/START criteria are criteria for rational prescribing in older people developed in 2008 [14] comprising two lists; the “STOPP” list of PIPs, and the “START” list of PIOs. In the STOPP/START tool, the authors structured their criteria according to physiological systems to mirror the usual organisation of drug formularies. The propositions were validated using an 18-member panel Delphi consensus where agreement was determined by the kappa-statistic for agreement and participants were able to suggest additional criteria if desired [15].

The POPI tool was developed in part by the Delphi consensus method. Prior to the Delphi consensus process, the authors compiled a list of possible propositions.

The authors structured POPI around 100 propositions classified according to biological systems and divided into omissions and inappropriate prescriptions. The number of propositions was chosen as “a good compromise between the number of major biological system to explore, the number of items in the geriatric lists and the maximum number of items compatible with a tool easy use” [8] (p. 2). The authors then compiled a list of health problems frequently encountered in paediatric practice, according to frequency in the general population (source not specified), prevalence (derived from data from the French National Insurance Fund for Employers for long term conditions), and frequency as cause for hospitalisation (per French hospital medico-administrative records). The authors identified health problems from this list, referred to as “themes”, that would either require drug intervention or where pharmacological intervention would be considered inappropriate.

For each selected theme, the authors conducted a literature search to identify recommendations on management. There was a requirement for recommendations to be evidence-based but the authors did not specify the level of evidence. Only recommendations published after 2000 were accepted and these were then weighted by date of publication. Accepted sources of recommendations were the French Health Products Safety Agency (Agence Française de Sécurité Sanitaire des Produits de Santé), the French National Authority for Health (Haute Autorité de Santé Française), the French Society for Paediatricians (Sociétè Française de Pédiatrie), the American Academy of Pediatrics (National Guideline Clearing House), and the National Institute for Health and Clinical Evidence (NICE) Cochrane Library [sic]. They also used the MEDLINE database to search for examples of medication error and inappropriate prescription (search strategy unpublished).

The propositions were then validated by a two-round Delphi consensus. Sixteen experts, including pharmacists and paediatricians, were included, of whom ten responded to both rounds. The process of recruitment of experts is not described, only that most pharmacists were members of the French Society of Clinical Pharmacy and most paediatricians were members of the French Society of Pediatricians. 

Initially, 108 propositions were presented to the experts. There is inconsistency between the 2014 (English) compared with the 2016 (French) publication in that the 2014 paper states 104 criteria were validated, whereas the 2016 paper states 101 criteria were validated. Furthermore, while it is stated in the 2014 paper that 104 propositions were validated, in fact 105 propositions are included in their final list. The difference between the two papers is due to three instances of combining two statements into a single criterion, and the omission of one proposition in the 2016 (French) publication. Specifically, two propositions about desmopressin for nocturnal enuresis are combined into one and six propositions about atopic eczema are combined into four. A proposition about benzyl benzoate for scabies is omitted. Other than this, the described process and criteria are the same.

Two propositions were removed following the consensus study due to new contraindications having been published for the use of these drugs in children, therefore 102 propositions were ultimately validated. In the 2016 French language publication describing the consensus validation of the POPI tool [11], the authors state that 101 of 108 criteria were validated. For the purposes of evaluation and discussion below, the English language published list of 105 validated propositions in the 2011 report is used.

### 3.4. Development of the Modified Popi UK) Tool

The modified POPI (UK) tool was developed in order to apply the POPI tool, which was based on a mixture of French, UK, and US guidelines, to UK practice [12]. A number of medications identified in the original POPI criteria are either not in usual use or unavailable in the UK, while some criteria directly conflicted with national UK clinical guidelines.

Each of the 105 criteria in the English language publication of the POPI tool were compared to relevant UK clinical guidelines from NICE, the Scottish Intercollegiate Guideline Network (SIGN), and the British National Formulary for Children (BNFc). In cases where there were no relevant guidelines or directly contradictory guidelines, criteria were removed. If guidelines differed, criteria were modified to reflect UK guidelines.

In comparison to the original criteria, 49 criteria were not changed. 29 were modified to meet UK guidelines, four criteria were combined into two, and 23 were omitted altogether. Omitted criteria included the removal of the clinical category of “Mosquitos”, which comprised seven criteria. 

### 3.5. Development of the PIPc

Like the POPI tool, the PIPc was developed by a two-round Delphi consensus method. Initial propositions were selected via a systematic literature search for previously developed indicators for paediatric prescribing.

Inclusion criteria were:Describe prescribing that is hazardous or known to be ineffectiveDescribe prescribing in keeping with best practice/current guidelinesApply to the population of children < 16 years

Exclusion criteria were:Medications unavailable in the study settingCriteria that could not be applied in the absence of clinical informationCriteria containing medications with a low prevalence of use

A steering group of academic and clinical general practitioners (GPs), academic and clinical pharmacists, a pharmacoepidemiologist/statistician, and a postdoctoral researcher, assessed each criterion. Those that were not felt to meet the above conditions were excluded. In some cases, criteria were modified to meet the need to be applicable without access to clinical information, for example when evaluating data from a dispensing database. 

The panel for the Delphi consensus comprised eighteen specialists, nine from the Republic of Ireland (three GPs, three paediatricians, and three pharmacists) and nine from the UK (three GPs, three paediatricians, and three pharmacists). 

The two-round Delphi consensus resulted in twelve criteria being accepted into the final PIPc.

### 3.6. Characteristics of the Tools

The use of the POPI tool is not specifically limited to any particular clinical setting. The propositions were selected from paediatric health problems in the general population and as causes for hospitalisation, suggesting the tool would be relevant to both primary and secondary care. However, no primary care specialists or general practitioners were involved in the development of the tool. The paediatric population is not explicitly defined by the authors but some propositions are age-specific, for instance pharmacological treatment for attention deficit disorder is described as inappropriate “before age six (before school)” [8] (p. 7) and topical 0.1% tacrolimus is considered inappropriate for atopic eczema “before 16 years of age” [8] (p. 6).

The criteria of the POPI tool cover a wide range of the aspects of rational prescribing, including all three categories of underprescribing, overprescribing, and misprescribing. Underprescribing errors are specifically identified in the tool as omissions. The inappropriate prescription propositions include some examples of overprescribing and misprescribing.

The modified POPI (UK) tool shares the characteristics of the original POPI tool. 

The PIPc is a tool that has been designed specifically for application in primary care settings. Unlike the POPI tool, it has been developed to be applicable without access to clinical information, meaning that it can be used to evaluate data from large previously collected prescribing databases where clinical information is often either omitted or concealed.

The authors of the PIPc define their paediatric population as children under 16 years of age. The age at which young people transition from paediatric to adult healthcare services can vary depending on health needs, social circumstances such as attendance in full-time education, and availability of specialist services [16]. In addition to the population specified for the tool, some criteria further specify particular age ranges, for example, “Loperamide should not be prescribed to children under 4 years” [11] (p. 8).

The PIPc criteria describe almost as broad a range of types of potentially irrational prescribing as POPI despite having far fewer criteria. The only aspects of irrational prescribing not contained within the PIPc that are covered by POPI are misprescribing of dosage, duration, and duplication.

### 3.7. Validation studies

The POPI tool has been evaluated in both a clinical validation [17] and repeatability study [18] (Table 3). The very high rate of PIPs in the community pharmacy is not further analysed in the published report. 

The published repeatability study of the POPI tool [18] found good repeatability despite the high complexity and mixed implicit and explicit approach of the tool.

The POPI tool was developed using French, American and UK guidelines and has been validated in clinical practice, with the above study showing that it is able to detect some potentially irrational prescribing in French settings. It is not yet known whether it detects irrational prescribing that correlates to adverse events or patient outcomes, or whether it could be used to evaluate prescribing outside French practice.

Although not published as a clinical validation study, there is a published study using the PIPc criteria to detect potentially irrational prescribing [19] (Table 3). In this study, the criteria were applied to a national pharmacy claims database in Ireland, the Primary Care Reimbursement Service (PCRS) with a cross-sectional methodology. The database records pharmacy claims for medicines for eligible patients prescribed by general practitioners or transcribed from hospital prescriptions by general practitioners, with limited patient demographic data (age, gender and region). No clinical details of the patients are recorded. The rate of PIO rose to 11.5% when including the criterion relating to co-prescription of a space device. Similarly, a single criterion had a large impact on PIPs and when this criterion, relating to carbocisteine, was removed the PIP rate fell to 0.29%.

One significant limitation in this study was that it highlighted the difficulty in applying even the intentionally simple and explicit criteria of PIPc to retrospective anonymised data. The age of patients in the PCRS database was recording in age bands of 0–4 years, 5–11 years, and 12–15 years. In several cases, these bands overlapped age limits described in PIPc criteria. In order to analyse the data, the authors made calculations to estimate the number of children of a certain age. For example, to calculate the number of children under 2 years in the 0–4 years band, the total number of children in the band was divided by 5 and multiplied by 2. This assumes a normal distribution of ages, which may not be the case.

There is no published repeatability study of PIPc.

A clinical validation study of the modified POPI (UK) tool has only been published in abstract form [12] (Table 3). There is currently no published repeatability study for the modified POPI (UK) tool.

### 3.8. Comparison with Existing Adult Rational Prescribing Tools

All of the paediatric tools identified cover a range of types of rational prescribing, with the POPI and modified POPI (UK) tools covering a particularly broad range. By comparison, the majority of adult tools identified by Kaufmann et al. [4] had a narrower focus. 

Of the 46 adult tools identified by Kaufmann et al., the median number of aspects of rational prescribing covered by each tool was 4.5, similar to the PIPc, which covers five aspects. However, four tools did cover eight or nine categories of prescribing, which is comparable to the POPI and modified POPI (UK), which cover eight.

The PIPc is designed for use in primary care settings, while the POPI tool and modified POPI (UK) are developed for application in a range of settings. The breadth of applications of both the POPI tool and modified POPI (UK) tool is similar to a number of rational prescribing tools for older adults, which have been used in settings including nursing homes, emergency departments, and primary care.

Of the adult tools evaluated by Kaufmann et al., the majority of tools (28) were explicit, a minority (8) were implicit, and the remaining 10 used a mixed approach like POPI. Implicit criteria may be more accurate, as they can take into account individual patient requirements, but this may come at the cost of reliability as they are more dependent on the rater’s knowledge and judgement [5]. The reverse is true of explicit tools, which are less reliant on rater judgement and therefore might be expected to have greater repeatability and reliability and be less time-consuming to apply, with concomitant lower accuracy. Therefore, mixed tools may stand to inherit both the advantages and disadvantages of each approach.

The authors of the original POPI state that the tool comprises explicit criteria; however, in both the POPI and modified POPI (UK) tools a number of criteria contain judgement-based and patient-specific considerations. Other propositions require taking into account the patient’s co-morbidities and entire medication regimen, characteristics that are usually considered components of implicit (patient-specific) criteria. For example, several propositions require the rater to make subjective judgements, such as in the theme of Attention Deficit Disorder, which includes “Pharmacological treatment before age six…except in severe cases” [7] (p. 7).

In some cases, the modified POPI (UK) amended criteria replace subjective judgements with explicit quantified cut-offs to reflect similarly precise recommendations in the UK guidelines. For instance where the original POPI tool lists a PIO of “Asthma inhaler appropriate for the child’s age”, the modified POPI (UK) criterion reads, “Asthma inhaler appropriate for the child’s age (aged < 5 years, either Metered Dose Inhaler with spacer system or nebuliser; age 3–5 years Dry Powder Inhaler may be appropriate)” [12].

The PIPc criteria are entirely explicit and do not require evaluation of a patient’s condition or subjective judgements. One criterion that may require non-anonymised data for full evaluation, however, is the recommendation in the Respiratory System theme, that “Children under 12 years who are prescribed a pressurised metered-dose inhaler should also be prescribed a spacer device at least every 12 months” [11] (p. 8). In order to assess this with certainty, the rater would need to be able to see all prescriptions for the child within the prior twelve months. Nonetheless, these criteria are all explicit.

## 4. Discussion

This systematic literature review identified three rational prescribing tools for use in paediatric practice, the PIPc, the POPI tool, and the modified POPI (UK) tool.

There are a number of research and clinical applications for rational prescribing tools.

The varying characteristics of the three paediatric rational prescribing tools identified have implications for their use and impact in future work. PIPc is intended only for primary care settings, while the POPI and modified (UK) POPI tools can be applied in any paediatric setting. 

All three tools could be used for both clinical governance and research purposes to identify areas of problematic prescribing, compare rates of irrational prescribing between settings, grades or specialties of prescribers, or regions. Because the tools provide a means to quantify rational prescribing, they may also facilitate the evaluation of educational or quality improvement interventions. The tools could also be used to assess factors associated with problematic prescribing.

In terms of structure and complexity, the POPI tool comprises a relatively high number of criteria compared with many other tools, although there is one published tool targeted at older adults with 392 quality indicators (not all of which relate to rational prescribing), ACOVE-3 [20]. The PIPc has closer to the lowest number of criteria of the adult tools. Some of the tools detailed in the Kaufmann systematic review have as few as ten criteria, for instance the Medication Appropriateness Index (MAI) [21], which is also targeted at older adults. A simple count of criteria is not necessarily a useful measure of complexity however. For example, in the case of the MAI, it is intended that all ten criteria are applied to each drug a patient is prescribed, where some systems simply list medications that are contraindicated or essential.

The high number of propositions and mixed implicit and explicit approach of the POPI tool makes it quite high in complexity, thus it requires a high level of clinical knowledge to apply. Some patients may fall within multiple themes, for instance Pain and fever might be expected alongside a number of other themes with an infectious focus, such as Urinary Infections and ENT Infections, and other themes describe long-term conditions that any child might have as co-morbidity. The theme of Vitamin Supplements and Antibiotic Prophylaxis includes a proposition describing minimum vitamin D intake, which would need to be assessed for every child. This would therefore require a high level of familiarity with the tool for accurate use and necessitates access to a high level of information about each patient.

By contrast, the PIPc is by design a tool that is simpler to apply and that requires minimal clinical information about a patient. The only clinical diagnosis specified in the tool is a presumed diagnosis of asthma in two criteria, which it appears that a rater is intended to presume on the basis of the prescriptions described, e.g., “An inhaled short-acting beta-2 agonist should be prescribed to children under 5 years who are also taking a leukotriene receptor antagonist for presumed asthma” [11] (p. 8). The PIPc is likely to be quicker to apply and does not require the high level of clinical information required by the POPI tool. However, it is also less broad and therefore it will not identify some aspects of irrational prescribing such as duplication, inappropriate drug duration, or incorrect drug dosage. There are also fewer clinical conditions included within the PIPc as compared with POPI, which may not reduce its efficacy as a screening tool in general settings but might reduce its usefulness in more specialist settings.

Further work comparing the sensitivity of the tools to detect rational prescribing and their utility in different clinical settings would be informative. In addition, it would be valuable to assess whether higher rates of irrational prescribing as detected by the tools is associated with poorer clinical outcomes or increased rates of adverse drug events.

As electronic prescribing becomes increasingly widespread, algorithmic clinical decision support systems have been developed to help alert clinicians to potentially inappropriate prescribing. In the older adult population where the Beers criteria and STOPP/START criteria are well-established, a computerised clinical decision support system integrating these rational prescribing tools has been developed [22]. This may be another avenue for further development towards greater rational prescribing for children by integrating one or more of the identified rational prescribing tools in a similar model.

The study of rational prescribing in children is a neglected area of research [23,24]. Studies of the value of these tools in different clinical settings by different investigators is needed to evaluate how useful the tools are. Such studies are essential to improve rational prescribing in different paediatric populations.

## 5. Conclusions

This systematic literature search identified three rational prescribing tools for use in assessing potentially inappropriate prescribing in paediatric settings, the PIPc, the POPI tool, and the modified POPI (UK) tool. We have outlined the characteristics of the tools, including their modes of design, aspects of rational prescribing assessed, and intended practice settings, which may assist readers in making use of the tools in their own clinical practice or for further research. The paucity of paediatric rational prescribing tools compared to adult tools shows that this remains a relatively underdeveloped field of study with great potential for future research.

## Figures and Tables

**Figure 1 ijerph-17-01473-f001:**
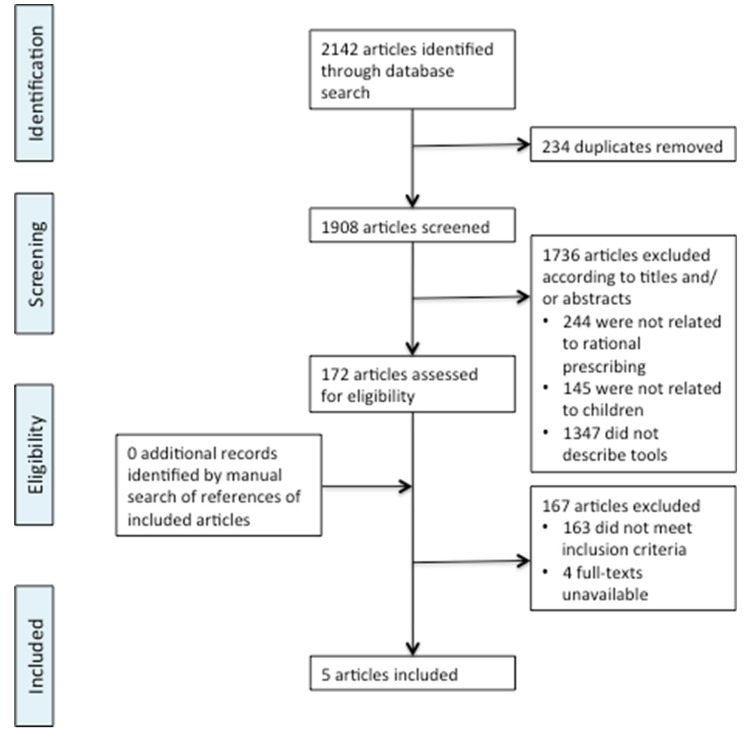
Flowchart of screening process for literature search. Adapted from the PRISMA group statement [8].

**Table 1 ijerph-17-01473-t001:** Results of systematic literature search.

Authors	Title	Year of Publication	Country	Name of Rational Prescribing Tool	Number of Criteria
Prot-Labarthe et al. [8]	POPI: A tool to identify potentially inappropriate prescribing practices for children (French).	2011	France	POPI	9 (partial list)
Prot-Labarthe et al. [9]	POPI (Pediatrics: Omission of Prescriptions and Inappropriate prescriptions): development of a tool to identify inappropriate prescribing.	2014	France	POPI	105
Prot-Labarthe et al. [10]	Consensus validation of a tool to identify inappropriate prescribing in pediatrics (POPI) (French).	2016	France	POPI	101
Corrick et al. [11]	Developing paediatric rational prescribing criteria.	2017	UK	Modified (UK) POPI	80
Barry et al. [12]	PIPc study: development of indicators of potentially inappropriate prescribing in children (PIPc) in primary care using a modified Delphi technique.	2016	Ireland and UK	PIPc	12

**Table 2 ijerph-17-01473-t002:** Characteristics of the paediatric rational prescribing tools.

Rational Prescribing Tool	Development Method	Healthcare Setting	Patient Group	Aspects of Inappropriate Prescribing								
				Misprescribing								
				Drug Choice	Dosage	Duration	Duplication	Drug-Disease Interaction	Drug-Drug Interaction	Drug-Food Interaction	Overprescribing	Underprescribing
POPI (Pediatrics: Omissions of Prescriptions and Inappropriate Prescriptions), Prot-Labarthe et al. [9,10,11]	Delphi consensus	Not specified	Children	♦	♦	♦	♦	♦	♦		♦	♦
Modified POPI (UK), Corrick et al. [8]	Modification of prior tool	Not specified	Children	♦	♦	♦	♦	♦	♦		♦	♦
PIPc (indicators of potentially inappropriate prescribing in children), Barry et al. [13]	Delphi consensus	Primary care	Children < 16 years	♦				♦	♦		♦	♦

**Table 3 ijerph-17-01473-t003:** Validation or repeatability studies.

Tool	Setting	Number of Children	Prevalence of PIPS	Prevalence of PIOs	Reference
POPI	Emergency department	15973	3.3%	2.6%	17
	Community	2225	26.4%	2.6%	
POPI	Emergency department	20	N/A (repeatability study)	N/A (repeatability study)	18
PIPc	Primary care	414,856	3.5%	2.5%	19
POPI(UK)	Emergency department and inpatient	400	32 in total		20

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
