# Peer review of "Paediatric Rational Prescribing: A Systematic Review of Assessment Tools"

_ijerph, 2020, doi:10.3390/ijerph17051473_

Round 1

Reviewer 1 Report

This is a systematic review of prescribing tools utilized in pediatric care. The manuscript is well written, the consensus methodology is applied  appropriately, the abstract is an excellent synopsis of the study, the discussion section is thorough, and conclusions follow from the results.

Since the authors found that PIPc contained more elegant and explicit criteria, they might speculate on the development of automated clinical decision support tools that could be applied in assessing rational prescribing in children. Its inclusion in the discussion is optional, but would add further insight into how prescribing rules might be computerized, especially considering advancements and trends in digital health and pharmacogenomics. In addition, because many medications used in children need to be compounded from other dose forms, prescribing rules must be robust enough to include these situations.

This submission advances our understanding of the current state of prescribing tools available to assess prescribing for children. Well done!

Author Response

Many thanks for your time and input you have given to reviewing our work. Please find below the changes we have made in response to your feedback. A brief discussion of electronic clinical support systems using rational prescribing criteria has been added to the discussion. More in-depth discussion of digital prescribing would be very interesting but was felt to lie outside the remit of this systematic review.

Reviewer 2 Report

Dear authors,

thank you for your work od review on the pediatric rational prescribing topic. I hope the following comments will help you in advancing and finalizing your work.

INTRO

An introduction to the topic lacks. Please, provide the definition of "rational prescribing" and a description of the literature on the topic, by explaining why it is important, as you wrote at the beginning - line 30.

Please, add some lines at the end of the introduction, to describe why you are studying this topic, what are the potential implications you expect from these tools (i.e. for medical training and education, for children and patients behaviors) and so on.

METHODS

Please, add a paragraph 2.2 on the aspect on which is focalized the analysis of the papers and follow them in describing the results.

RESULTS

A detailed description of the screening process lacks. The Figure 1 contains more information rather than the text: please, provide more details.

I suggest reviewing the results with a more critical approach, by following the additional paragraph 2.2 as I suggested above. A table or a box with bullet points or similar, to synthesize results (not only the description of the tools), will be useful to the readers.

DISCUSSIONS

Please, add the implications you can draw from your findings.

Author Response

Many thanks for your time and input you have given to reviewing our work. Please find below the changes we have made in response to your feedback.

A definition and explanation of the importance of rational prescribing has been added to the introduction.

Discussion of the implications of the work was added to the Discussions section rather than both introduction and discussion, in the interests of concision and in accordance with another reviewer's feedback.

Further detail was added to the methodology section regarding analysis of the papers. Full narrative description of the detailed screening flowchart in Figure 1 was felt to be repetitive and was not added into the text.

A table was added to the results to synthesise the resulting rational prescribing tools identified through the search. A brief explanation was added to precede Table 2 to explain the analysis of the tools.

Reviewer 3 Report

Thank you for the submission of your article to this journal.

There is a need to make changes as follows:

In the methods section, you need to recognise what type of literature review has been performed. If this is a systematic review (as mentioned in the abtarct), it should repeated here and the review process should be supported using methodological citaions.

The exact date of the review should be stated. 

Did you use the Boolean search?

More details are need regarding who has done what in the search process, full-text appraisal, gray literature, manual search etc. Please use the PRISMA checklist and fill out gaps in the presentaion style.

The begining part of the results should provide details on the demographic details of the studies selected for data analysis, country, year, aim, etc.

In table 1, what does the bullet points mean?

Tabulation of findings would help with reduction of lenght writings. There is no need to provide all details from the selected studies, but you can use tables to summarise such details, highlight similarities and differences, and focus on the findings in connection to your review aim. 

It seems that you have repeated your review findings in the discussion section. Please be precise and just discuss your findings in terms of implications of the findings for education, research, and practice, without repeating the findings section. 

Author Response

Many thanks for your time and input you have given to reviewing our work. Please find below the changes we have made in response to your feedback.

Systematic literature search was added to the methods. The date of the search was clarified.

Description of manual search of bibliography was made clearer in the methods as well as a brief description of the process of analysing the papers. The authors are satisfied that all points on the PRISMA checklist relevant to systematic review without meta-analysis have been addressed.

Boolean search terms were used as identified in the italicised search terms stated.

A table summarising findings with demographics has been added to help readers understand and navigate the results. Details of the selected studies have been included as the purpose of the review was to find and describe the extant rational prescribing tools for readers’ use.

Repetition of results has been removed from the discussion section. Additional discussion of the implications of the characteristics of the tools to inform their further development or use within future work has been included.

An explanation of Table 1 (now Table 2) has been added to the text.

Author Response

Many thanks for your time and input you have given to reviewing our work. Please find below the changes we have made in response to your feedback.

Attention deficit disorder: this heading was used as the original POPI used the term “Attention deficit disorder with and without hyperactivity” whereas in the UK “Attention deficit hyperactivity disorder” is the usual term in children; the discussion was comparing the two tools with these differing headings. An ellipsis was added to the italicised heading to show it abbreviates the full heading from the original POPI.

The detail regarding numbers and identity of reviewers has not been changed due to requests from other reviews for increasing the detail in the methodology and adherence to guidance around transparency of the roles of authors.

The discussion of the qualities of the tools has been moved from the Discussion to Results section and a repetitive section removed.

Further discussion of the impact of the work has been added to the Discussion

Round 2

Reviewer 3 Report

There are misleadings in the presentation of Your findings. Why there are many discussion in the results section without appropriate citation? from pages 4-10 you have provided a lot of descriptions With a few citaions for paragraphs. In the previous review round, I said that there is no need to copying and pasting the findings of the selected studies, but you should use tables and figures to provide details in terms of similatities and differences between the studies. 

Did you aim to develop an instrument based on the reviewed Tools? This is unclear in the aim. 

Again, the discussion is very weak and include a few citations and without comparison of findings with those of other studies. 

Author Response

We are sorry that the reviewer does not feel that we have tried to answer their comments.

We had previously added an extra table as suggested. We also had to respond to the comments of the three other reviewers.

We have added a sentence at the end of the Introduction clarifying and expanding the aim as requested.

We have added another additional table (Table 3) comparing the findings from the papers.

We have correspondingly deleted parts of the text for the Results.

Discussion – we have expanded as suggested.